# Surgical Phase Recognition in Inguinal Hernia Repair—AI-Based Confirmatory Baseline and Exploration of Competitive Models

**DOI:** 10.3390/bioengineering10060654

**Published:** 2023-05-27

**Authors:** Chengbo Zang, Mehmet Kerem Turkcan, Sanjeev Narasimhan, Yuqing Cao, Kaan Yarali, Zixuan Xiang, Skyler Szot, Feroz Ahmad, Sarah Choksi, Daniel P. Bitner, Filippo Filicori, Zoran Kostic

**Affiliations:** 1Department of Electrical Engineering, Columbia University, New York, NY 10027, USA; cz2678@columbia.edu (C.Z.); mkt2126@columbia.edu (M.K.T.); yc3998@columbia.edu (Y.C.); ky2446@columbia.edu (K.Y.); zx2366@columbia.edu (Z.X.); sls2305@columbia.edu (S.S.); 2Department of Computer Science, Columbia University, New York, NY 10027, USA; sn3007@columbia.edu (S.N.); fa2581@columbia.edu (F.A.); 3Intraoperative Performance Analytics Laboratory (IPAL), Lenox Hill Hospital, New York, NY 10021, USA; dbitner@northwell.edu (D.P.B.); ffilicori@northwell.edu (F.F.); 4Zucker School of Medicine at Hofstra/Northwell Health, Hempstead, NY 11549, USA

**Keywords:** surgical phase recognition, inguinal hernia repair, robotic-assisted laparoscopic surgery, computer vision, deep learning, AI, convolutional neural network, transformers

## Abstract

Video-recorded robotic-assisted surgeries allow the use of automated computer vision and artificial intelligence/deep learning methods for quality assessment and workflow analysis in surgical phase recognition. We considered a dataset of 209 videos of robotic-assisted laparoscopic inguinal hernia repair (RALIHR) collected from 8 surgeons, defined rigorous ground-truth annotation rules, then pre-processed and annotated the videos. We deployed seven deep learning models to establish the baseline accuracy for surgical phase recognition and explored four advanced architectures. For rapid execution of the studies, we initially engaged three dozen MS-level engineering students in a competitive classroom setting, followed by focused research. We unified the data processing pipeline in a confirmatory study, and explored a number of scenarios which differ in how the DL networks were trained and evaluated. For the scenario with 21 validation videos of all surgeons, the Video Swin Transformer model achieved ~0.85 validation accuracy, and the Perceiver IO model achieved ~0.84. Our studies affirm the necessity of close collaborative research between medical experts and engineers for developing automated surgical phase recognition models deployable in clinical settings.

## 1. Introduction

Artificial intelligence (AI) in the form of machine learning (ML) or deep learning (DL) refers to training machines to automatically perform a selection of “intelligent tasks”. The potential advantages of AI in medicine include labor reduction and, possibly, improvement of the quality of analysis for healthcare [1]. In surgery, the abundance of video data from intraoperative recordings creates an opportunity for training computer programs to interpret visual data using computer vision (CV) and DL [2,3,4]. Although CV/DL can in principle be used in many facets of video interpretation, such as instrument recognition and even skill assessment, the most widespread use is currently in surgical phase recognition [2,3].

Surgical phase recognition is of interest for its potential utility in workflow analysis and evaluation of technical quality. Automation of the assessment of operative phases would be a useful adjunct in such analysis for reducing costs and labor. Several types of operations are amenable to segmentation/classification by DL models. Most reports have focused on laparoscopic surgeries, especially sleeve gastrectomy [5,6], sigmoidectomy [7], myotomy [8], and cholecystectomy [9]. One of the most prevailing benchmarks is based on the Cholec80 [9] surgical workflow dataset. The earliest attempt was by the dataset developers in 2016, achieving an accuracy of 75.2%. The model, namely EndoNet, adopted a convolutional neural network (CNN) for feature extraction, followed by a support vector machine and a hidden Markov model for temporal phase inference. Later reports incorporated time dependency into the DL models in multiple fashions, with some examples including SV-RCNet (81.6%, 2018) using LSTM [10] and TeCNO (88.6%, 2020) using 3D convolutions [11]. Utilization of the attention mechanisms was reported by TMRNet (90.1%, 2021) [12] and Trans-SVNet (90.3%, 2021) [13]. A recent attempt in 2023 using 3D convolution with positional encodings reported 92.3%, which is believed to be the best result on Cholec80 [14].

Generally, studies on surgical phase recognition have been single-institutional studies, with internal resources of both: (1) video acquisition and labeling and (2) CV/DL algorithm design and validation [5,6,7,8]. Recently, more papers on DL have been featured in clinical journals, which indicates that CV/DL methods in surgery are moving from pure demonstrations of “Can we do it?” to more applied uses, answering: “What can we do with it?” [4]. Such a shift in emphasis requires ongoing collaborations between surgeons and engineers. Aside from collaboration per se, additional advantages might be found in the introduction of competition between groups, yielding repeatable, more robust results and novel solutions. This was carried out with the Surgical Action Triplet Recognition Challenge hosted by the Medical Image Computing and Computer-Assisted Interventions (MICCAI) 2021, in which groups competed to produce the best possible algorithm to predict so-called triplet classes (instrument + action + target) during laparoscopic cholecystectomy [15]. In the CholectTriplet 2021 challenge, the winning group outperformed the baseline model published by the hosts [15,16].

The studies reported in this paper describe the establishment of a team and a process for collaboration between clinicians and engineers working on applications of CV/DL in surgeries, development of robust ground-truth annotations for RALIHR phase recognition, verification of baseline recognition accuracies, and exploration of advanced methods for improving the performance. The studies incorporated the training of three dozen students who, in a competitive fashion, produced a large set of easily comparable results. The key outcomes of the studies are a robust workflow and CV/DL models with accuracies in the range of ~0.85 for a non-ideal dataset acquired from eight surgeons.

## 2. Materials and Methods

### 2.1. Video Acquisition, Annotation, and Processing

A large video dataset of RALIHR is available at our health system (Northwell Health, Hempstead, NY, USA) through commercial agreement with a video recording device platform (C-SATS, Inc., Seattle, WA, USA). Videos of RALIHR obtained from eight surgeons were downloaded and stripped of all potentially identifying information prior to CV/DL processing. This study was deemed exempt by the Northwell IRB (IRB #19-0254).

An iterative, collaborative method and approach to ground-truth annotations for the RALIHR phases was developed and implemented. Phases were defined, with the priority being feasibility of machine-based detection. For example, periods of transitions or pauses between clearly defined phases commonly exist in real-life surgeries. Regarding them as a part of adjacent phases would negatively affect model learning due to the injection of ambiguous visual information. Here, a compromise was made between strict surgical definitions and the ease of task completion, hedging toward stricter definitions of each of the phases. All annotations were completed by a single surgical trainee (D.B.). The surgical phases to be annotated are defined in Table A1.

Videos were uniformly processed in the following manner. De-identified videos were processed with the FFmpeg video/audio processing library to reduce the frame-per-second (FPS) from 30 FPS, as encoded on the C-SATS platform, to 1 FPS to reduce the computational load. This reduction has a negligible impact on the phase recognition task. A frame resolution of either 760 × 468 or 1280 × 760 was maintained, the audio track was removed, and the video format was kept in H.264. Completely de-identified videos were shared with the engineering team via upload to the institutional Google Drive folder (Google, Mountain View, CA, USA).

### 2.2. Competitive Model Creation

The simplest possible way of formulating video-based surgical phase recognition is to treat it as a classification problem, where the frames from the videos are individually categorized into a set of predefined phases. This modeling approach easily adapts to different choices regarding temporal information: the frames are either independent of one another or temporally related, depending on the model configuration. A variety of image and video classification methodologies can be chosen for this task. For an image classification model, the input is processed on a per-frame basis. A video-based model, on the contrary, takes input of short video clips, configurable in duration.

We tasked a class of master-level students with creating their own CV/DL models for RALIHR phase recognition, or a customized model published in the literature, for rapid development and experiments of state-of-the-art models in surgical phase recognition. Students were divided into groups and provided with de-identified video data and the ground-truth annotations. Background literature on CV/DL models for surgical video analysis was suggested. No restrictions were placed on how or with which resources students could devise their models. The results had to be submitted into a competition on Kaggle, which is a web environment featuring various open-source datasets (Kaggle, San Francisco, CA, USA). The teams then produced an accompanying report with code provided in GitHub repositories (GitHub, San Francisco, CA, USA). The main metric of interest is the accuracy score for multi-class classification. The projects were completed in the Spring Semester of 2022 as a part of a class at Columbia University.

### 2.3. Confirmatory Baseline Study

In order to explore the various methodologies attempted by different student groups, a confirmatory baseline study was conducted to verify and establish a comprehensive procedure for data processing and model evaluation. Statistics about the dataset were analyzed with respect to the frequency and duration of every surgical phase in each video. A severely imbalanced data percentage between phases was identified as one of the major factors reducing the model performance. We addressed this by merging data-scarce phases into data-abundant ones that share enough visual similarity. Seven of the original fourteen phases were preserved, as shown in Table A2. To reduce overfitting and promote model generalization, every input image was augmented by random resized cropping (taking a randomly distorted portion of the image with a ratio of 0.9–1.1 and resizing the output to a common shape of 224 × 224) and a random horizontal flip with a probability of 0.5. The effect of diversity in the training set was also explored.

For the purpose of validating the pipelines before exploring advanced architectures, the DL models in this study were chosen with the aim of being simple enough to verify the robustness of other components of the pipeline and to minimize the influence on the final results imposed by the model itself. The ResNet50 backbone [17] was selected for the baseline studies due to its robustness and ability of adaptation to both frame-based and temporal models. Residual neural networks (ResNet) are a popular backbone type of convolutional neural networks (CNN) that have achieved great success in general image classification tasks [18]. Convolution has been a widely used technique in signal processing due to its ability to extract meaningful low-dimensional features from large redundant data sources (e.g., videos and audio). The features can further be used to perform tasks such as classification. The ResNet-50 model is named for having 50 layers in total with internal skip connections, known as the residuals. It is a rather efficient but naïve architecture since all input images are considered independent of each other. This simplification makes the model well-suited for a baseline study. A wide variety of advanced models also adopt ResNet as a feature extraction backbone.

We explored a number of scenarios using different train–test splits of the whole dataset, detailed in Section 3.1. They were proposed with considerations of: (1) data availability at the time of study, (2) consistency for comparisons between cases, and (3) diversity of data sources from different surgeons. The optimal train–test split found by the experiments involved all collected videos into the dataset. The goal was to keep as many videos as possible for model training, while preserving enough quantity and diversity for evaluation (detailed in Section 3.3.2). This particular split was adopted by later experiments with advanced models. The study was completed in the Summer Semester of 2022.

### 2.4. Explorations of Advanced Models

Following the established baselines from Summer 2022, several new models with more advanced architectures were explored by the engineering team, with ResNet-50 used for comparisons. All models were trained by minimizing the cross-entropy loss with a learning rate of 1 × 10^−5^. The study was conducted during the Fall Semester of 2022 and the Spring Semester of 2023.

Perceiver IO [19]: Transformers [20] were initially introduced to the field of natural language processing (NLP). They are founded on the attention mechanism, which successfully addresses long-term memory issues by globally combining the information from the entire input sequence. Perceiver IO is a modification of the naïve transformers and enables the transformer architecture to handle different data modalities without changing the model structure. Input data are first processed by a pretrained ResNet-50 that converts images into representative features. The features are then fed to the attention modules either in the form of independent frames or video clips per configuration. ResNet-50 weights are frozen during model fine-tuning.

Video Swin Transformer [21]: The initial architecture of the Swin Transformer [22] is a hierarchical transformer model that uses shifted windowing for attention computation to greatly improve the model efficiency. The Video Swin Transformer generalizes this idea to the video domain, where 3D local windows are shifted instead of 2D windows used for images. This allows the model to learn long-range dependencies across multiple frames. The Video Swin Transformer outperforms previous state-of-the-art models on several video classification tasks while using fewer parameters and less computation. We adopted the tiny variant (Swin-T) of the Video Swin Transformer model pretrained for action recognition tasks [23] in non-medical scenarios, modified the final classification layer, and fine-tuned it on our dataset. The input images are directly fed into the model in short video clips and are passed through several transformer blocks. Features are extracted by each block using a 3D window of size 2 × 4 × 4. Similar to a CNN-like structure, spatial sizes are gradually down-sampled, and the dimensions of features are gradually increased. Readers are referred to the Video Swin Transformer paper for more details [21].

## 3. Results

The studies of competitive model creation, confirmatory baseline, and advanced model explorations were carried out in a sequential manner, each motivated by previous results and conclusions.

### 3.1. Dataset and Annotations

From the total dataset of the 211 videos collected from 8 surgeons, 2 were excluded for having inconsistent data vs. annotations, while the remaining 209 videos were accordingly incorporated into the dataset: 186 videos from surgeon “01” and 23 videos from the 7 other surgeons. Heterogeneous components of the videos included additional phases (primary hernia repair, an optional suturing task to close the defect, primarily) in 20 cases, incomplete phases (video beginning after the start of preperitoneal dissection) in 2 cases, and early mesh excision (mesh placed before the scoring or dissection of the peritoneum) in 2 cases. The recorded videos in the dataset have an average length of ~54 min, each containing ~33 recurring phases (same phases may appear more than once at different times during a surgery). Different train–test dataset splitting schemes were explored at different stages throughout the study. Some representative cases are presented in Table 1, the details of which are described in the following sections.

### 3.2. Competition Results

Twelve student groups submitted their models to the competition as part of the assignment. We used the train–test setup in Case 1 (Table 1), including all the videos that were available at that time of the course. The models’ main performances and structures are summarized in Table 2.

Seven out of the twelve student groups adopted a recurrent Long Short-Term Memory network (LSTM) into their architectures for temporal embeddings, while another two approached this goal using 3D convolutions [11,24]. The increase of accuracy compared before and after adding LSTMs was generally reported to be ~1%. The other three groups merely used a CNN and did not incorporate such an embedding, but two of them (Groups 3 and 7) postprocessed the model outputs by taking the average of the predictions within a sliding window of thirteen frames. The smoothing technique reported a 1.9% and 1.0% increase in accuracy for Groups 3 and 7, respectively, surpassing the performance of adding an LSTM. The two top-performing groups, with accuracies of 0.8199 and 0.8111, adopted the Temporal Memory Relation Network (TMRNet) [12]. The architecture features both CNN and LSTM appended with an attention layer for aggregation of long-range temporal dependencies.

Regarding complexity–accuracy trade-offs, it is worth noting that the output averaging is simple to compute, and attention mechanisms are shown to be more potent in long-term memory, while being fully parallelizable, as opposed to recurrent networks. We therefore argue that output-smoothing and attention mechanisms are mutually diverse but more appropriate candidates than LSTMs for integrating temporal surgical information.

### 3.3. Pre-Processing and Evaluation Pipelines

While the competition models were developed independently by different student groups using various data processing and validation pipelines, the engineering team requires the establishment of a robust pipeline on which all future models can be evaluated and uniformly compared. The following subsections of the paper discuss the proposed solutions for several pressing issues generally reported by the student teams.

#### 3.3.1. Phase Merging

Visualizations of dataset statistics are provided in Figure 1. A brighter color indicates longer durations (e.g., peritoneal closure and reduction of hernia appears drastically more often than adhesiolysis and catheter insertion). While the annotation scheme is strict and intuitive with more phase definitions, a deep learning model can easily suffer from a lack of training data on less frequent categories. The competition models uniformly reported low class-wise accuracies on: (1) blurry, with <0.1% of total frames, (2) primary hernia repair, with ~0.7% of total frames, and (3) stationary idle, with ~1.0% of total frames. A merging strategy was adopted which proposed to take all phases with less than 3% frequency and combine them into other more predominant phases that are visually similar to the original ones (Table A2). Exceptionally, two less frequent phases (out of body and peritoneal scoring) were kept due to having distinct visual features.

Sufficient training of a ResNet-50 model, on the same Case 1 (Table 1) dataset before and after merging, revealed a ~0.25 increase in the macro F1-score, from 0.5349 to 0.7851, while yielding a similar accuracy of 0.7862 and 0.7870. Considering that the learning from an imbalanced dataset can be a topic in-and-of-itself, phase merging has been presented to be a decent work-around for this particular study. All experiments and analyses described below were conducted with merged labels.

#### 3.3.2. Data Diversity

Apart from the imbalance between surgical phases, there also existed an imbalance in data sources, i.e., there was a dramatically larger number of videos from surgeon “01” (186 videos) compared to the 7 other surgeons (23 videos in total). We further conducted a 5-fold cross-validation using Case 3 (Table 1) with 40 videos, with all surgeons mixed as evenly as possible (17 videos randomly sampled from surgeon “01”, 23 videos taken from the 7 other surgeons). The average accuracy yielded by the cross-validation was 0.6916. As a comparison, training with more videos of a single surgeon, as in Case 4 (Table 1), resulted in a much lower accuracy of 0.4808. This informs that training on a variety of surgeons is a better option even when few videos are available.

We further explored Case 5 (Table 1) with all available data included while mixing up different surgeons for more diversity. A total of 188 videos were sampled, with 173 of them from surgeon “01” and 15 from the other surgeons for the training set, leaving 21 videos, with 15 from surgeon “01” and 6 from the others, for the validation set. In order to spare as many videos as possible for model training, especially the ones from the “other” surgeons, we deliberately did not construct a test set and kept a small but diverse validation set: at least one video from every surgeon was included in the validation set, except for surgeon “08”, who had only a single recorded video. The results on a sufficiently trained ResNet-50 are summarized in Table 3, emphasizing the difference in performances between different surgeon groups.

With this particular train–test splitting scheme, we were able to increase the accuracy on other surgeons by more than 20% compared to Case 4 and achieved better accuracy than the cross-validation, while roughly maintaining the best result on surgeon “01”. Therefore, Case 5 (Table 1) was adopted into the exploration of new models in Section 3.4.

#### 3.3.3. Edge Cropping

Some recorded videos contain text boxes with the surgical robot status (as shown on the Figure 2a) overlaid on the image. This raises concern that important classification information may undesirably leak into the DL models for these particular videos, causing models to learn from the text rather than from the surgical content. To assess if this problem may occur, we utilized a method called Grad-CAM [25,26], which traces the gradients of the model output backwards and projects them to the image input. The projected results represent a heat map of the regions of interest from which the predictions are generated by a DL model. The result is exemplified in Figure 2 using a sufficiently trained ResNet-50 on Case 5 (Table 1). The image shows the reduction of the hernia phase with the model focusing on the hernia defect (upper part of the image). A brighter color indicates higher importance of that region, illustrated in pixel values.

With 1000 randomly sampled images from the dataset, the average gradient on the position with text boxes was reported to be 0.23 (±0.01), while the average gradient from the pristine areas of the images was 0.31 (±0.07). An independent *T*-test comparing the two areas resulted in a score of 4.91 (*p*-value < 1 × 10^−10^). This finding eliminated previous concerns by providing quantitative evidence that the model infers mainly from regions outside of the text box areas, yielding no specific cropping steps.

### 3.4. Other Advanced Models

After establishing a comprehensive pipeline with the baseline study, we further experimented with more advanced model architectures using Case 5 (Table 1). The ability to embed temporal information was emphasized as a comparison to ResNet-50, which assumes independence between input images. Both the selected models, Video Swin Transformer and Perceiver IO, have been reported as strong candidates for video modeling [19,21]. Table 4 summarizes their results obtained from the validation set.

Specifically, Perceiver IO adopted a latent space of size 8 × 512 and an input clip of 16 frames. Performance was boosted by as much as ~7%, to 0.8414, without any fine-tuning on the ResNet-50 backbone, compared to the ~1% increase induced by LSTMs (described in Section 3.2), demonstrating the superiority of transformers in learning long-term temporal dependencies. Swin-T, on the other hand, takes an input clip of 10 frames divided into 3D patches of shape 2 × 4 × 4. It therefore contains a smaller number of parameters comparable to that of ResNet-50, however it reported the best performance of 0.8491. It was indicated that a more powerful feature extraction backbone (e.g., Swin Transformer or ViT [27]) may further improve the final outcome.

An illustration of Video Swin Transformer predictions on 10 videos from the validation set compared to the ground truths is shown in Figure 3. The confusion matrix on the entire validation set is presented in Figure 4. A darker cell color represents more samples, labeled by its proportion within the validation set. The model achieved both high precision and recall even on the less frequent phases, such as out of body and peritoneal scoring. However, minor confusions were observed between particular phases. An example is that ~39% of frames from preperitoneal dissection were classified into reduction of hernia. This behavior was anticipated as we noticed that the borderline between these two phases can sometimes be vague due to the nature of phase definitions (detailed in Table A1). Transitionary idle is another easily misclassified phase, considering that it occupies the time period between other well-defined phases, and thus could possibly share their visual features.

## 4. Discussion

We presented a series of studies conducted in a collaborative fashion by the medical team and the engineering team. We built a comprehensive pipeline incorporating data collection, annotation, data processing, DL-based model creation, and evaluation for surgical phase recognition for RALIHR.

### 4.1. Ground-Truth Annotations

A major amount of effort and time were dedicated to ground-truth annotations. There are significant challenges with annotations for CV/DL projects in surgery, the critical one being the balancing between the availability of annotators and the requisite expertise for doing the work properly (e.g., college students cannot simply take up annotation without knowing basic surgical techniques) [28,29]. Even with surgical trainees or attendings, the annotations must be very clear and unambiguous. The annotation rules have been iteratively refined to align the expert surgical knowledge with pragmatic engineering needs.

### 4.2. Classroom Competition

The engineering skills required to create and modify CV/DL models to generate meaningful output for a particular goal demand notable expertise. We sought an effective way for our academic collaboration and developed the idea of a competition amongst the MS-level students to test this approach, with appropriate training and faculty guidance. The results indicate that the approach is feasible and highly effective in the early stage of such projects, as the entirety of this collaboration took only several weeks in the Spring of 2022, with seven different DL models tested and compared for clarity of operative phases and in-between video scenes.

It should be remarked that DL models experimented in such a competitive setting are often taken off-the-shelf without being carefully tuned for the intended dataset. High heterogeneity exists between student groups in terms of their approaches to data processing and model evaluation. We emphasize the necessity of confirmatory study as an immediate follow-up to the competitions. No further attempts should be made on performance improvements if there is not much certainty that the methodology to be explored is the desirable or even correct direction to pursue.

### 4.3. CV/DL Methodologies

Experiments with more than a dozen DL models with various architectures were conducted throughout the study. It was evident that different architectural components yielded inherently different performances. A general conclusion is that temporal embeddings are crucial to model accuracy at the higher-end. Image-based models tend to reach their bottleneck at around ~0.8, including Tiny-ViT [27], which reported an accuracy of 0.8271 despite being a particularly strong image classification backbone. Despite the overall success of video-based models over image-based ones, there were particular cases where such models failed. The competition results (Table 2) suggested that X3D [24], a traditional video recognition model using 3D convolutions, may be unsuitable for surgical phase recognition. However, its variant TeCNO was able to reach satisfactory results both in the literature [11] and on our dataset (Table 2). We also experimented with MoviNet [30], which uses a similar 3D convolution mechanism. The model obtained an accuracy of 0.8121, which was inferior to other advanced models. In comparison, attention-based models such as Perceiver IO and Video Swin Transformer (including the TMRNet reported in competition, Table 2) performed exceptionally well among the experimented architectures, beating other models based on LSTMs and 3D convolutions.

It was also reported by the baseline study that the impact of the augmentation technique on the ResNet-50 results was not significant. Subsequent experiments of removing random horizontal flipping improved the accuracy in the advanced models, possibly indicating that the horizontal alignment of visual elements in RALIHR (such as surgical tools or the view perspective of operative areas) plays an important role in phase prediction. We also studied the impact of color- and lighting-based augmentations, including random color jitter and histogram equalization. We observed the best results when applying contrast-limited adaptive histogram equalization (CLAHE), an improvement of the naïve histogram equalization technique that improves contrast in local regions of the input image. The underlying rationale is to brighten the darker parts in a recorded video as they may also contain critical features for phase recognition.

One missing part is finding an appropriate way to encode prior information. Surgical procedures tend to bear hierarchical structures which can be easily defined by humans but hard to learn by DL models. For example, a suturing operation would never precede a dissection/scoring operation in a fully recorded video. This type of knowledge can either be adopted as a postprocessing technique independent of the model, as in SV-RCNet [10], or incorporated into the loss calculation that supervises and facilitates model learning. There can be a wide range of possible approaches to be experimented in future studies.

### 4.4. Clinical Applications

The ultimate goal of our collaborative studies is to bring the CV/DL models into operating rooms. This goal requires the model to robustly generalize to unseen videos from unseen surgeons, while being able to process input data and make inferences fast enough in real time.

The generalizability, to date, is mainly limited by the quantity of annotated data. A multi-institutional study with 36 different surgeons on DL-based phase recognition for laparoscopic cholecystectomy found that differences in model performance between surgeons were minimal [31]. Our findings suggest differently, which implies that the conclusions from laparoscopic cholecystectomy may be limited to surgeons with small case volumes. Additionally, the technical performance of RALIHR may be less standardized than a cholecystectomy, and systematic differences between surgeons may matter more for CV/DL models in these cases.

Another difficulty regularly encountered was the exceedingly long time a model takes to train. A single epoch may run for 2.5 h on the 188-video training set for complex architectures such as the Video Swin Transformer, even when using an advanced graphics processing unit (GPU) such as the NVIDIA A100. The inference time for these models will also be consequently longer (as described in Table 4). Furthermore, it may be impossible during the deployment in operating rooms to gain access to high-end GPUs, making it important to maintain a feasible hardware requirement when exploring new models.

The current model, with future advancements, could be integrated into hospital and operating room software to help optimize the surgical workflow. Operating room staff can be continuously informed about the status of the procedure and know when a surgeon requires different instruments or needs help. For example, if a certain phase is taking longer than it should, an alarm system could be linked to the software to indicate to supervisors that the surgeon may require assistance. It can also help operating room staff to know what instruments will be needed during a surgery so that they are ready when it is time for the next phase. Automatic phase recognition in real-time has important implications in optimizing the organization of the operating room and possibly improving the quality of the care delivered.

## 5. Conclusions

The focus of the presented work was the automation of the analysis of recorded RALIHR surgeries. We described a series of studies on CV/DL models for surgical phase recognition, executed by an inter-institutional team of medical experts, engineering students, and researchers. The studies encompassed data collection, image preprocessing, ground-truth annotations, and algorithm development. More than a dozen models were evaluated and compared for multiple-use case scenarios, through classroom competition and focused research, achieving accuracies in phase recognition as high as ~0.85. This is the first comprehensive study comparing the performances of DL models for phase recognition in RALIHR. The studies are unique in that they effectively engaged students who participated in an advanced-level course in the process of comprehensive investigation on many models. This collaboration between the medical team and the engineering team is essential to optimizing operating room workflow through innovative computer vision and deep learning methodologies.

## Figures and Tables

**Figure 1 bioengineering-10-00654-f001:**
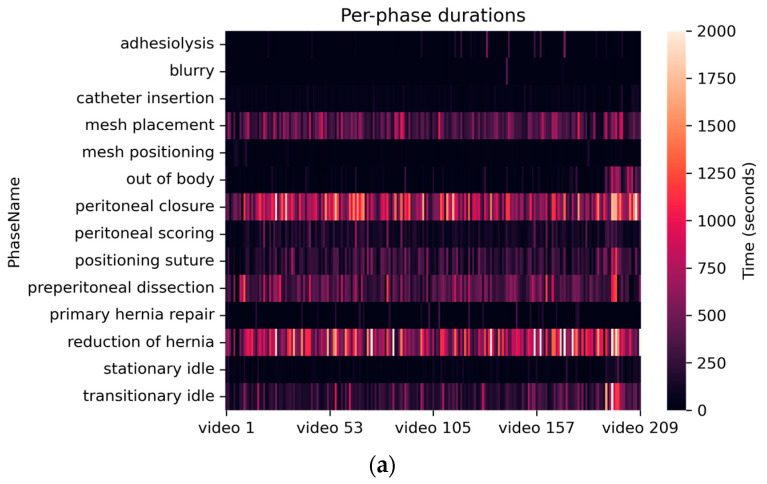
Heat plot of durations (i.e., number of samples) of each surgical phase in the entire dataset with 209 videos: (**a**) before phase merging and (**b**) after phase merging.

**Figure 2 bioengineering-10-00654-f002:**
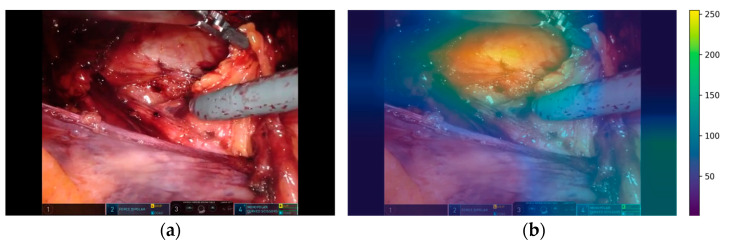
Example of region of interest (**b**) overlaid on the original image (**a**) from which the predictions were generated.

**Figure 3 bioengineering-10-00654-f003:**
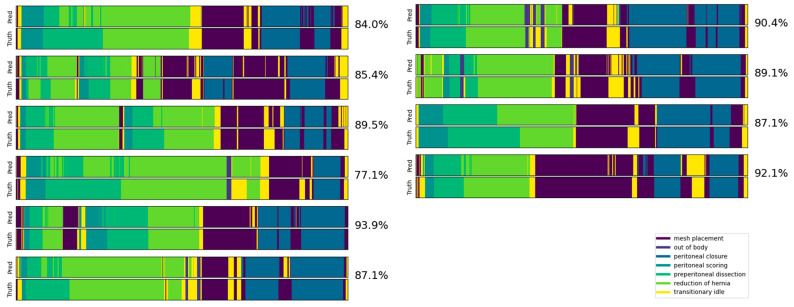
Example of predictions (top) vs. ground truths (bottom) from the top-performing Video Swin Transformer model for 10 sample videos from the validation set. The accuracy of each video is labeled on the right.

**Figure 4 bioengineering-10-00654-f004:**
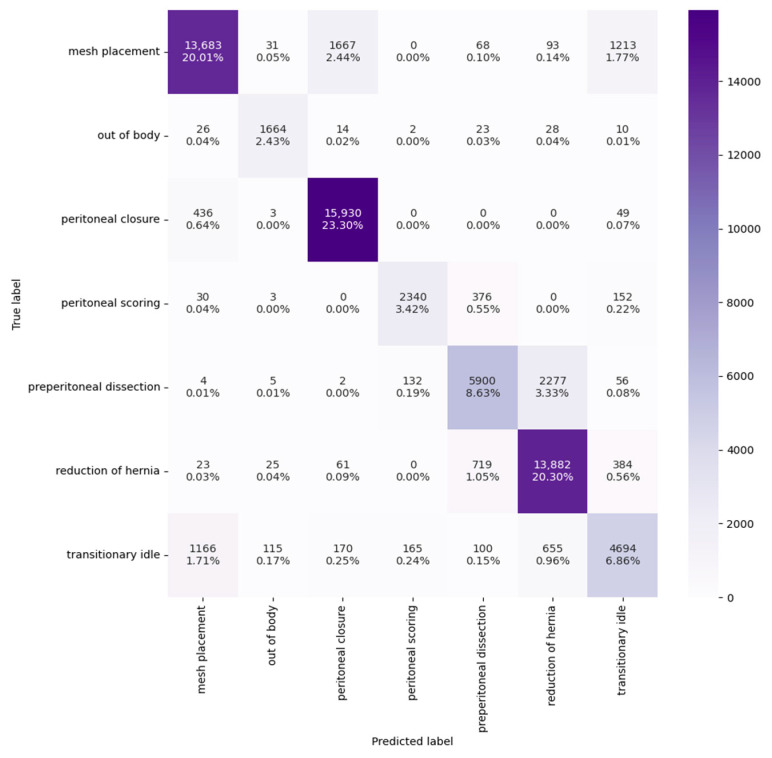
Confusion matrix of the top-performing Video Swin Transformer model. The columns correspond to the model predictions and the rows are the ground truths.

**Table 1 bioengineering-10-00654-t001:** Different train–test splits (in number of videos) used in the study. The test/validation accuracies on a sufficiently trained ResNet-50 (with the phase-merging technique described in Section 3.3.1) are provided for reference.

Case	Total #Videos	Train		Test		ResNet Accuracy
		Surgeon 01	Others	Surgeon 01	Others	
1	120	70	-	47	3	0.7870
2	186	136	-	50	-	0.8015
3	40 *	17	23	-	-	0.6916
4	209	186	-	-	23	0.4808
5	209	173	15	15	6	0.7704

* Case intended for cross-validation without a test set. Accuracy is represented by the average of the best-performing epoch on each fold.

**Table 2 bioengineering-10-00654-t002:** Accuracies of competition models and their architectures from Spring 2022.

ID	Accuracy	Model	Architecture
1	0.8199	TMRNet	CNN + LSTM + Attention
2	0.8111	TMRNet	CNN + LSTM + Attention
3	0.7955	MobileNet	CNN + Output Smoothing
4	0.7951	TeCNO	CNN 3D
5	0.7948	TMRNet	CNN + LSTM + Attention
6	0.7930	-	CNN + LSTM
7	0.7917	EfficientNet	CNN + Output Smoothing
8	0.7816	SV-RCNet	CNN + LSTM
9	0.7809	-	CNN + LSTM
10	0.7659	ConvNeXt	CNN
11	0.7619	-	CNN + LSTM
12	0.1006	X3D	CNN 3D

**Table 3 bioengineering-10-00654-t003:** Validation accuracy of ResNet-50 on Case 5 (Table 1).

Source	Accuracy	Videos
Surgeon 01	0.8096	15
Surgeons 02–08	0.7025	6
All surgeons	0.7704	21

**Table 4 bioengineering-10-00654-t004:** Performance of the explored models.

Model	Accuracy	Clip Length	Parameters (M)	Inference Time * (ms)
ResNet-50	0.7704	1	25.6	8.7 ± 0.7
Perceiver IO	0.8414	16	36.3	47.4 ± 0.5
Swin-T	0.8491	10	28.0	13.14 ± 5.2

* Inference time includes only the model forward pass tested with 300 samples on NVIDIA A100.

## Data Availability

Data cannot be made publicly available due to privacy issues.

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
