# Peer review of "Surgical Phase Recognition in Inguinal Hernia Repair—AI-Based Confirmatory Baseline and Exploration of Competitive Models"

_bioengineering, 2023, doi:10.3390/bioengineering10060654_

Round 1

Reviewer 1 Report

I am really grateful for being given this opportunity as a reviewer for the article, entitled “Surgical Phase Recognition in Inguinal Hernia Repair – AI-Based Confirmatory Baseline and Exploration of Competitive Models”.

The report by Zang and colleagues is very intriguing, describing the necessity of close collaborative research between medical experts and engineers for developing automated surgical phase recognition models deployable in clinical settings.

The authors have tried to improve the surgical performance using robotic-assisted laparoscopic inguinal hernia repair (RALIHR) surgeries.

The study is really unique as the authors suggest in the text that this is the first comprehensive study comparing the performances of deep learning (DL) models for phase recognition in RALIHR and that they effectively engaged students who participated in an advanced-level course in the process of comprehensive investigation on many models.

The conclusions by the authors are consistent with the evidence and arguments presented.                          

I hope as they suggest that collaboration between the medical team and the engineering team is further encouraged for optimizing operating room workflow through innovative computer vision and deep learning methodologies.

Thank you.

Author Response

Response to Reviewer 1.

Thank you for the evaluation of the manuscript, which indicates that you are satisfied with the content and the presentation.

Reviewer 2 Report

This is an excellent review paper that summarizes key points in detail, and is of considerable interest to surgeon physician-scientists interested in quality improvement and real-time assessment of surgical outcomes and performance. 

A few questions that the authors could consider commenting on:

(1) Was port placement and robotic positioning similar in all cases reviewed, or was the model able to accommodate / assess port / camera / instrument placement in all cases? 

(2) Is performance able to be assessed? In terms of patient outcomes / LOS etc?

(3) Was type of suture / stitch used evaluated or standardised among the evaluation population?

(4) Were reasons that certain lengths of time took longer than normal able to identified and assessed? This could be of interest to surgeons - for exam was a hernia unable to be fully reduced or was the size of the defect variable?

No concerns about quality of English language. Well done.

Author Response

Response to reviewer 2.

Thank you for the evaluation of the manuscript, which indicates that you are satisfied with the content and the presentation.

Reviewer 3 Report

well done and interesting paper to read and to know about 

Author Response

Response to reviewer 3.

Thank you for the evaluation of the manuscript, which indicates that you are satisfied with the content and the presentation.